# The amyloid-beta forming tripeptide cleavage mechanism of γ-secretase

David M Bolduc*, Daniel R Montagna, Matthew C Seghers, Michael S Wolfe*, Dennis J Selkoe*

Ann Romney Center for Neurologic Diseases, Brigham and Women's Hospital, Harvard Medical School, Boston, United States

**Abstract** γ-secretase is responsible for the proteolysis of amyloid precursor protein (APP) into short, aggregation-prone amyloid-beta (Aβ) peptides, which are centrally implicated in the pathogenesis of Alzheimer's disease (AD). Despite considerable interest in developing γ-secretase targeting therapeutics for the treatment of AD, the precise mechanism by which γ-secretase produces Aβ has remained elusive. Herein, we demonstrate that γ-secretase catalysis is driven by the stabilization of an enzyme-substrate scission complex via three distinct amino-acid-binding pockets in the enzyme's active site, providing the mechanism by which γ-secretase preferentially cleaves APP in three amino acid increments. Substrate occupancy of these three pockets occurs after initial substrate binding but precedes catalysis, suggesting a conformational change in substrate may be required for cleavage. We uncover and exploit substrate cleavage preferences dictated by these three pockets to investigate the mechanism by which familial Alzheimer's disease mutations within APP increase the production of pathogenic Aβ species.

*For correspondence: davidmbolduc@gmail.com (DMB); mwolfe@rics.bwh.harvard.edu (MSW); dselkoe@partners.org (DJS)

Competing interests: The authors declare that no competing interests exist.

## Introduction

Alzheimer's disease (AD) is the most common form of dementia and currently the sixth leading cause of death in the United States, with no disease-modifying therapeutics available. A central and pathological hallmark of AD is the deposition of amyloid-beta (Aβ) plaques in the brain (*Hardy and Selkoe, 2002*). These plaques are comprised of aggregates of Aβ peptides, which are formed by the sequential cleavage of the membrane embedded amyloid precursor protein (APP) by two proteases—β-secretase first removes the ectodomain of APP, then γ-secretase cleaves the remaining C-terminal fragment within its transmembrane domain (TMD) to liberate Aβ from cellular membranes. Via its proteolytic component presenilin (*Li et al., 2000*; *Wolfe et al., 1999*), γ-secretase processes the TMD of APP into Aβ peptides of differing lengths, mostly producing a more benign Aβ species 40 amino acids in length, termed Aβ40, as well as lesser amounts of a longer, more aggregation prone and pathogenic species, Aβ42. The total amount of Aβ42, as well as the ratio of Aβ42/40, are thought to be key mediators of Aβ pathogenesis, a hypothesis strongly supported by the fact that nearly all the 200-plus autosomal-dominant familial Alzheimer's disease (FAD) mutations in presenilin-1, -2 and APP increase the Aβ42/40 ratio (see www.alzforum.org/mutations).

Due to the strong link between γ-secretase catalyzed Aβ formation and AD pathogenesis, the development of γ-secretase targeting therapeutics has been of high interest over the past two decades (*Golde et al., 2013*; *De Strooper, 2014*). Both γ-secretase inhibitors (GSIs) and a γ-secretase modulator (GSM), the latter working by an unknown mechanism to influence γ-secretase to produce shorter, presumably less pathogenic Aβ species, have failed in recent clinical trials. The failure of GSIs is due at least in part to toxicities from cleavage inhibition of other γ-secretase substrates such as Notch (*Doody et al., 2013*; *Golde et al., 2013*). Little is known about how γ-secretase recognizes the transmembrane domain of substrates, given that no consensus amino acid cleavage motif has

**eLife digest** Individuals with Alzheimer's disease generally have deposits known as "amyloid plaques" in the brain. These plaques are made up of a mixture of molecules called amyloid beta peptides that clump together and are thought to be a key cause of the disease. The amyloid beta peptides vary in size; the larger peptides tend to be more prone to forming clumps than the smaller ones and are thus more toxic to the brain.

An enzyme called gamma-secretase makes amyloid beta peptides by cutting up a protein called APP. Proteins are made of chains of building blocks called amino acids and studies using a technique called mass spectrometry show that gamma-secretase cuts APP in segments of three amino acids at a time. The size of the amyloid beta peptides produced is determined by the positions in APP that gamma-secretase selects to cut. Therefore, understanding how the enzyme works could provide new opportunities for developing drugs to treat Alzheimer's disease.

Here, Bolduc et al. found that the human gamma-secretase enzyme has sites that amino acids in APP can bind to that help to guide the enzyme to cut APP by three amino acids at a time. These binding sites control where the enzyme cuts APP and therefore determines which amyloid peptides are produced. Previous studies have linked several naturally occurring mutations in the gene encoding APP to inherited forms of Alzheimer's disease. Bolduc et al. now reveal that several of these mutations affect the places that gamma-secretase cuts APP to produce amyloid peptides.

These findings may be helpful for developing drugs that could manipulate gamma-secretase to produce smaller, less harmful amyloid peptides. Gamma-secretase can cut many other proteins, and so a future challenge will be to find out if the enzyme cuts these other proteins in the same way that it cuts APP.

been identified for the more than one hundred γ-secretase substrates discovered to date (*Haapasalo and Kovacs, 2011*). Unfortunately, the further development of safe and effective γ-secretase targeting therapeutics has been held back by a fundamental lack of understanding of how γ-secretase recognizes and cleaves the TMDs of its many substrates, especially APP. Elucidation of this basic mechanism should at the very least add to our understanding of how Aβ is produced and may also aid in the development of safe and effective disease-modifying therapeutics.

Mass spectrometry studies have identified a complex mixture of products generated from γ-secretase's cleavage of the transmembrane domain of APP (*Matsumura et al., 2014*). Looking at the formation of these products over time, it is apparent that the TMD of APP is mostly processed via two major pathways. γ-secretase predominantly initiates endoproteolysis at a so-called *epsilon* (ε) cleavage site—after Leu49 or Thr48, generating Aβ49 or Aβ48 and two different APP intracellular domains (AICD), AICD 50–99 or AICD 49–99, respectively (*Kakuda et al., 2006*; *Sato et al., 2003*). Aβ49 and Aβ48 are then sequentially cleaved in increments of three amino acids to produce mostly Aβ40 and Aβ42, respectively (*Takami et al., 2009*). The two major pathways are therefore Aβ49 → Aβ46 → Aβ43 → Aβ40 and Aβ48 → Aβ45 → Aβ42 (*Figure 1A*) (*Fernandez et al., 2014*; *Takami et al., 2009*). There are, however, other Aβ species generated by γ-secretase through usually minor and sometimes overlapping, alternative pathways (*Matsumura et al., 2014*; *Olsson et al., 2014*). Importantly a shorter peptide, Aβ38, can be formed from both major pathways, originating from Aβ42 or Aβ43 (*Okochi et al., 2013*). Additionally, a third, sparingly used site of ε cleavage can lead to the production of Aβ47, which rather than being processed to Aβ44 is instead mostly cleaved to Aβ43, subsequently generating Aβ40 (Aβ47 → Aβ43 → Aβ40) (*Matsumura et al., 2014*).

Normally, γ-secretase uses the Aβ49 → Aβ40 and the Aβ48 → Aβ42 pathways to produce mostly Aβ40 and Aβ42 via a stepwise, tripeptide cleavage process. The mechanism that dictates this preferred tripeptide cleavage (and thus the driving force behind γ-secretase catalysis and Aβ formation) is completely unknown. In this study, we report that γ-secretase tripeptide cleavage is driven by three S' pockets within the active site of the enzyme. We identify specific substrate cleavage preferences dictated by the three S' pockets and exploit these preferences to determine the predominant mechanism of each FAD mutation within the transmembrane domain of APP, including a novel mechanism in which final cleavage products are uncoupled from initial ε pathway preference.

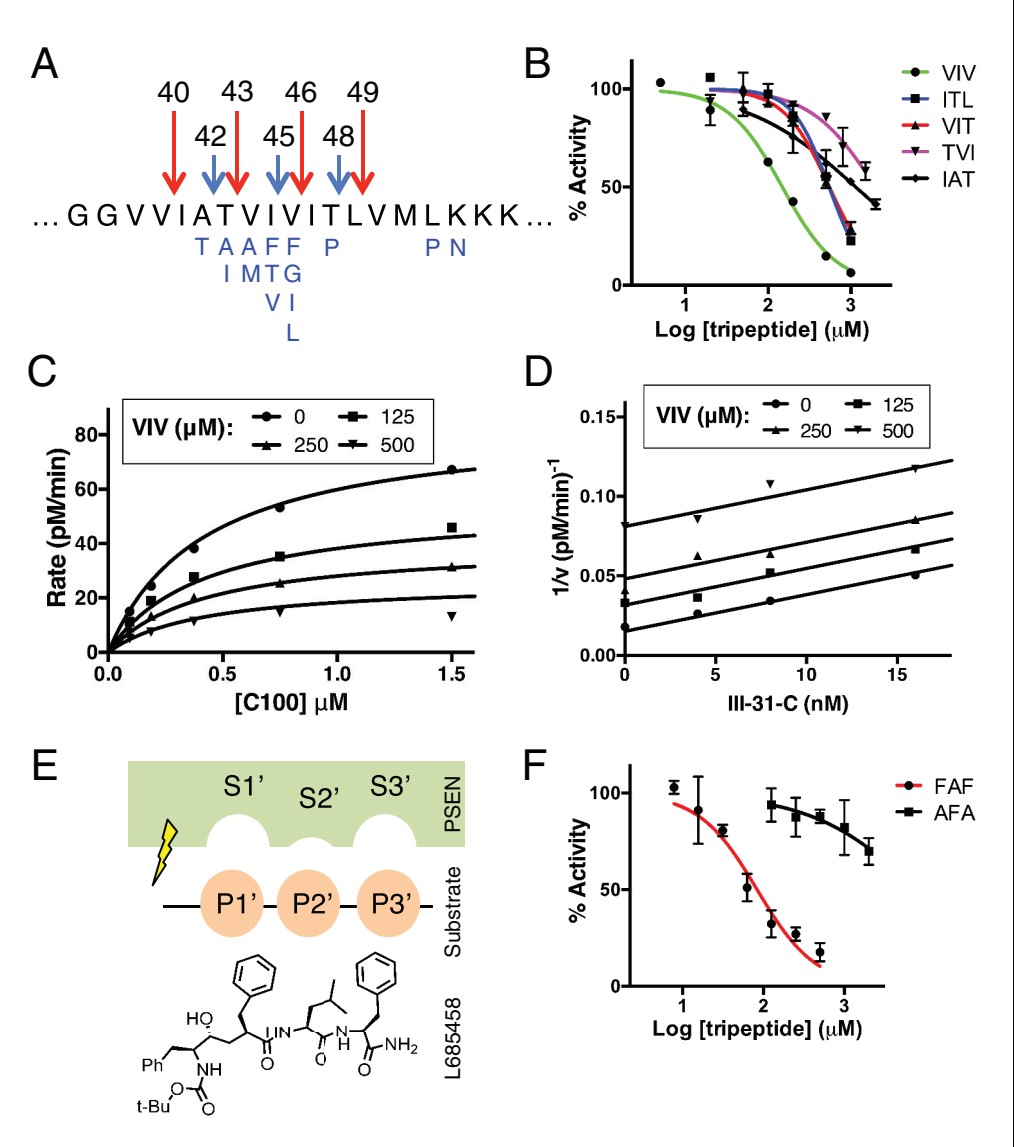

**Figure 1.** Tripeptide fragments of APP inhibit γ-secretase. (**A**) Schematic diagram of the major sequential cleavage pathways of the transmembrane domain of APP (Aβ49 → Aβ46 → Aβ43 → Aβ40 in red and Aβ48 → Aβ45 → Aβ42 in blue). Mutations causing Familial Alzheimer's disease are below the APP TMD in blue. (**B**) IC50 curves from the inhibition of γ-secretase activity by APP product tripeptide fragments. Mean ± SD, n = 2. (**C**) Noncompetitive inhibition of γ-secretase with VIV tripeptide, $R^2$ = 0.98. (**D**) Yonetani-Theorell plot for the mutually exclusive binding of VIV and the noncompetitive transition-state analog inhibitor III-31-C, $R^2$ = 0.98. (**E**) Cartoon representation of the three S' pockets of presenilin (PSEN) along with three P' amino acids of substrate and the transition-state analog L685,458. (**F**) IC50 curves from the inhibition of γ-secretase activity with FAF and AFA synthetic tripeptides. Mean ± SD, n = 2.

## Results

When studying enzyme catalysis much focus is appropriately placed on determining how an enzyme interacts with its substrate. However, oftentimes the manner in which an enzyme interacts with product (in the form of product inhibition) can be equally informative with regard to its catalytic mechanism. To this end, we asked whether the naturally produced tripeptide fragments of APP are inhibitors of γ-secretase. We found that all five tripeptides produced from the TMD of APP are indeed capable of inhibiting γ-secretase activity, albeit rather weakly with IC50 values ranging from

~150 µM to several mM (*Figure 1B*). Although these binding affinities are too low for the tripeptides to be involved in any form of biologically relevant feedback inhibition, we imagined the manner in which they inhibit γ-secretase could be instructive in elucidating the basic cleavage mechanism of the protease. We characterized the mode of inhibition of the most potent of the tripeptides, VIV, finding that these data fit well to a noncompetitive inhibition model, with a global $R^2$ of 0.98 (*Figure 1C*). Given that the tripeptide segments of the TMD of APP must occupy the active site of γ-secretase during catalysis, we hypothesized that VIV may compete for the same binding site on the enzyme as transition-state analogs. An inhibitor cross-competition analysis reveals this is likely true, with a series of parallel lines resulting from a Yonetani-Theorell plot demonstrating mutually exclusive binding of VIV and the transition-state analog III-31-C (*Figure 1D*).

We noticed that nearly all of the γ-secretase targeting transition-state analog inhibitors developed to date (e.g. L685,458, III-31-C) contain essentially a tripeptide fragment C-terminal to the transition-state-mimicking hydroxyl isostere (*Figure 1E*). Structure-activity relationship (SAR) studies have demonstrated that transition-state analog inhibitors containing only two amino acids here are relatively weak inhibitors compared to those comprised of three amino acids, while adding a fourth amino acid does not achieve additional potency (*Esler et al., 2004*). This suggests that there are three, and only three, putative S′ pockets in the presenilin active site that contribute to inhibitor binding. Of note, the SAR studies also suggest that while the putative S1′ and S3′ pockets are large and can accommodate amino acids of varying size, the S2′ pocket is small and inhibitors with an aromatic amino acid (phenylalanine) at this position have decreased potency by two orders of magnitude compared to those with less bulky aliphatic amino acids (*Esler et al., 2004*).

We imagined that the tripeptide APP TMD products may be binding these three putative S′ pockets to achieve inhibition. In agreement with this hypothesis, a synthetic tripeptide of the predicted optimal binding sequence (FAF, to fit in the large-small-large S1′-S2′-S3′ pockets) was a more potent inhibitor of γ-secretase activity than any of the naturally occurring APP-derived tripeptides, while a peptide predicted to clash with this binding site (AFA) is a very weak inhibitor, even at mM concentrations (*Figure 1F*).

Based on these results, we reasoned that the three putative S′ pockets within the γ-secretase active site are likely occupied by substrate prior to hydrolysis of the scissile bond as a means for γ-secretase to stabilize a transition-state-like scission complex with substrate. This would provide a simple mechanism for the preferred cleavage of APP in three amino acid increments, as well as provide an explanation for why γ-secretase mostly sticks to each major pathway, producing Aβ40 or Aβ42 after initiating cleavage of APP at either the L49 or T48 ε sites, respectively. Additionally, given that substrate movement in the form of helical unwinding is thought to be a required step in the poorly defined intramembrane protease cleavage mechanism (*Akiyama et al., 2015*; *Dickey et al., 2013*; *Fluhrer et al., 2012*; *Moin and Urban, 2012*; *Urban and Freeman, 2003*; *Ye et al., 2000*), the three S′ pockets could potentially provide a means for γ-secretase to stabilize its helical substrate in a more cleavable conformation, thereby lowering the activation energy required for catalysis.

To test this hypothesis, we took advantage of the fact that the putative S2′ pocket of γ-secretase is apparently small and has a reduced ability to accommodate a bulky amino acid such as phenylalanine. We predicted that we should be able to selectively decrease cleavage of either the Aβ49 → 40 pathway or the Aβ48 → 42 pathway simply by placing an aromatic amino acid at the P2′ position of APP at the initial ε cut site. In other words, an aromatic amino acid placed at V50 should reduce ε cleavage after T48, thereby lowering the amount of Aβ42 produced, thus decreasing the Aβ42/40 ratio. Conversely, an aromatic amino acid at M51 should reduce ε cleavage after L49, lowering the amount of Aβ40 produced and therefore increase the Aβ42/40 ratio (*Figure 2A*).

In an in vitro assay using purified γ-secretase to cleave recombinant C100-FLAG APP-based substrate, V50F and V50W both decreased the Aβ42/40 ratio, while the same substitutions at M51 increased the Aβ42/40 ratio as predicted (*Figure 2B*). MALDI/TOF mass spectrometry (MS) analysis of the corresponding AICD fragment revealed the complete elimination of AICD 49–99 for V50F and of AICD 50–99 for M51F (*Figure 2C*). Interestingly, in addition to the expected AICD fragments, both V50F and M51F are also cleaved to a minor extent after I47, producing AICD 48–99. The reason for this is unknown, although γ-secretase may be compensating for reduced cleavage through one of the two major pathways. Previous MS studies have demonstrated that the majority of Aβ47 is eventually processed to Aβ40, through an Aβ43 intermediate (*Matsumura et al., 2014*).



**Figure 2.** Selective blocking of the Aβ40 or Aβ42 pathways with aromatic amino acids placed at the P2′ position of ε cleavage. (**A**) Schematic diagram of the TMD of APP with pathway blocking aromatic amino acid mutations at the P2′ position for the T48 or L49 ε cleavage. (**B**) In vitro Aβ42/40 ratios with Phe or Trp mutations at V50 or M51. Aβ measured using Aβ40 and Aβ42 ELISA kits from Invitrogen. Mean ± SD, n = 3, t-test **<0.01, ****<0.0001. (**C**) MALDI/TOF MS of the AICD fragments generated from in vitro cleavage of C100: WT (AICD 50–99, expected mass: 6905.6, observed mass: 6907.4; AICD 49–99, expected mass: 7018.8, observed mass: 7021.3). V50F (AICD 50–99, expected mass: 6953.8, observed mass: 6949.8; AICD 48–99, expected mass: 7167.9, observed mass: 7163.5). M51F (AICD 49–99, expected mass: 7034.8, observed mass: 7030.1; AICD 48–99, expected mass: 7135.8, observed mass: 7131.8). (**D**) Aβ42/40 ratios measured from the media of HEK cells transfected with V50 mutants. Aβ levels measured by 6E10 ELISA. Mean ± SD, n = 3, t-test **<0.01, ***<0.001, ****<0.0001. (**E**) Aβ42/40 ratios measured from the media of HEK cells transfected with M51 mutants. Aβ levels measured by 6E10 ELISA. Mean ±

*Figure 2 continued on next page*

*Figure 2 continued*

SD, n = 3, t-test ***<0.001, ****<0.0001. (**F**) Aβ40 and Aβ42 levels for aromatic substitutions at V50 and M51 normalized to WT. Aβ levels measured by 6E10 ELISA. Mean ± SD, n = 3, t-test *<0.05, ***<0.001, ****<0.0001. (**G**) Total secreted Aβ levels (see Materials and methods) from the aromatic mutations at V50 and M51.

We obtained similar results measuring secreted Aβ after transiently transfecting HEK cells with full-length APP containing mutations at V50 or M51. Here, all mutations tested at V50 caused a reduction in the Aβ42/40 ratio, although none as robustly as the bulky aromatic amino acids Phe, Tyr and Trp (*Figure 2D*). And while mutations of smaller amino acids at M51 caused modest reductions in the Aβ42/40 ratio, aromatic substitutions here all caused substantial increases in the Aβ42/40 ratio (*Figure 2E*). As predicted by our model, there was a significant reduction in Aβ42 production for the V50 aromatic substitutions, with little change in the Aβ40 levels (*Figure 2F*). There was a similar predicted reduction in Aβ40 for the M51 aromatic mutants; however, we also see an increase in Aβ42 levels here (*Figure 2F*), likely because the normally less used Aβ48 → 42 pathway is compensating for the reduced flux through the preferred Aβ49 → 40 pathway, as these mutants all produced roughly equivalent amounts of total Aβ (*Figure 2G*).

Because each tripeptide cleavage event requires the reading of three amino acids of substrate at a time (as dictated by the three S' pockets), we reasoned we should be able to predictably shift the Aβ42/40 ratio by placing a Phe in the P2' position at each tripeptide cleavage event along the two major pathways (*Figure 3A*). As expected, Phe substitutions at V44 and I47 decreased the Aβ42/40 ratio, while Phe mutations at I45 and T48 increased the Aβ42/40 ratio (*Figure 3B and C*). The predicted Aβ42/40 shifts were nearly identical whether Aβ levels were measured from an in vitro assay (*Figure 3B*) or from a cell-based assay (*Figure 3C*). These results are of particular note, as I45F is a known FAD mutation (*Guerreiro et al., 2010*), likely indicating that the mechanism of this mutation is the Phe positioned in the P2' position at the Aβ43 cut site, blocking cleavage of APP through the less pathogenic Aβ49 → 40 pathway and favoring the pathogenic Aβ48 → 42 pathway. The Phe at I45 also lies in the S3' position for Aβ42 cleavage, likely making the precursor to Aβ42 production a better substrate for γ-secretase through a favorable P3'-S3' interaction. Mutating A42 to Phe completely blocked the formation of Aβ40 in agreement with our model (*Figure 3D*), while still allowing for the production of other Aβ species (*Figure 3—figure supplement 1*).

Although we expected the Phe substitutions at V44, I45, I47 and T48 to be acting independently of the pathway chosen at initial ε cleavage, an alternative explanation for the above results is that these mutations instead influence ε cleavage in favor of the final Aβ products measured. To investigate this possibility, we utilized an antibody that specifically recognizes the free N-terminus of AICD 50–99 (*Chávez-Gutiérrez et al., 2012*), and therefore the initiation of the Aβ49 → 40 pathway. Surprisingly, all four of these mutants actually caused a shift in ε cleavage toward the initiation of the opposite pathway. Although V44F and I47F cause reductions in the Aβ42/40 ratio, they shifted initial ε cleavage away from the Aβ49 → 40 pathway—nearly completely eliminating AICD 50–99. And while I45F and T48F increased the Aβ42/40 ratio, they shifted ε cleavage more in favor of AICD 50–99 compared to WT (*Figure 3E*). These AICD species were confirmed by mass spectrometry (*Figure 3F*).

To explore further the apparent ability of tripeptide cleavage preference to dissociate the normal connection between initial ε cleavage and final γ cleavages, we made double Phe mutants in which the ε cut site is controlled with a Phe at either V50 or M51, while placing a conflicting Phe at V44, I45, I47 or T48 in the initial pathway. Measuring Aβ secreted from transfected HEK cells, we clearly saw the double mutants behave almost identically to the single point mutants N-terminal to the ε cut site (*Figure 4A and B*). MS of the AICD fragments revealed the expected and complete blocking of AICD 49–99 or AICD 50–99 for the V50F and M51F containing double mutants, respectively (*Figure 4C*). Together these data suggest final γ cleavages can be completely uncoupled from initial ε cleavages.

Single Phe mutations at V46 and L49 both caused modest increases in the Aβ42/40 ratio. We predicted that since these mutations do not occupy the S2' pocket for either major pathway, V46F and L49F when paired with a neighboring Phe mutation should behave like its neighboring Phe mutant alone. Indeed, L49F-V50F had a reduced Aβ42/40 ratio compared to WT like V50F alone, while

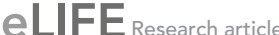

**Figure 3.** Phenylalanine mutations at P2' positions predictively shift the Aβ42/40 ratio. (**A**) Schematic diagram of Phe mutations at P2' positions at cut sites within the TMD of APP and the expected Aβ42/40 changes compared to WT. (**B**) In vitro Aβ42/40 ratios from γ-secretase cleavage of recombinant C100-FLAG substrates. Aβ measured using Aβ40 and Aβ42 ELISA kits from Invitrogen. Mean ± SD, n = 3, t-test *<0.05, **<0.01. (**C**) Aβ42/40 ratios from Aβ secreted from HEK cells. Aβ levels measured by 6E10 ELISA. Mean ± SD, n = 3, t-test **<0.01, ***<0.001, ****<0.0001. (**D**) Aβ40 levels measured from the conditioned media of HEK cells transfected with WT or A42F APP. Aβ levels measured by 6E10 ELISA. Mean ± SD, n = 3. (**E**) Western blot analysis of the AICD fragments generated from V44F, I45F, I47F and T48F in vitro. Total AICD was measured with anti-FLAG M2 antibody (green). Aβ49 → Aβ40 pathway preference was measured with an antibody specifically recognizing the N-terminus of AICD 50–99 fragment (red). (**F**) MALDI/TOF MS confirmation of the AICD fragments measured in (**E**): V44F (AICD 49–99, expected mass: 7018.8, observed mass: 7020.8), V45F (AICD 50–99, expected mass: 6905.6, observed mass: 6906.2;

*Figure 3 continued on next page*

*Figure 3 continued*

AICD 49–99, expected mass: 7018.8, observed mass 7020.1), I47F (AICD 49–99, expected mass: 7018.8, observed mass: 7019.5), T48F (AICD 50–99, expected mass: 6905.6, observed mass: 6907.4; AICD 49–99, expected mass: 7018.8, observed mass: 7019.5).

The following figure supplement is available for figure 3:

**Figure supplement 1.** Total Aβ from the A42F mutant.

T48F-L49F had an elevated Aβ42/40 ratio comparable to T48F (*Figure 4D*). Likewise, V46F-I47F displayed a reduced Aβ42/40 ratio similar to I47F alone, while I45F-V46F had a drastically increased Aβ42/40 ratio like I45F alone (*Figure 4D*).

To this point, aromatic amino acids placed in the P2′ position at each tripeptide cleavage site within the TMD of APP caused a predictable outcome without exception. We therefore reasoned that placing two phenylalanines in tandem, such that there is a Phe in the P2′ position at both major ε cut sites, should reduce overall cleavage. As predicted, a double mutant of V50F-M51F caused a sharp reduction in total AICD formation compared to WT and the V50F and M51F single point mutations alone (*Figure 5A*). However, AICD formation from V50F-M51F cleavage was not completely abolished; rather, its rate of production was markedly reduced (*Figure 5B*). This suggests γ-secretase has a means by which to overcome two aromatic amino acids in a row, which would seemingly conflict with its cleavage preferences.

The observation that both tripeptide cleavage products of APP and transition-state analog inhibitors are noncompetitive inhibitors of γ-secretase suggests that the subsites on γ-secretase for initial substrate binding and subsequent catalysis are spatially separate and distinct, meaning substrate movement is likely required after initial substrate binding but prior to catalysis. This would be in agreement with previous reports suggesting there may be an exosite on γ-secretase to which substrate initially binds prior to translocating to the active site for cleavage (*Kornilova et al., 2003*, *2005*). It would also agree with studies of other intramembrane cleaving proteases, which have proposed substrate movement in the form of helical unwinding is a prerequisite for catalysis (*Akiyama et al., 2015*; *Dickey et al., 2013*; *Fluhrer et al., 2012*; *Moin and Urban, 2012*; *Urban and Freeman, 2003*; *Ye et al., 2000*).

Given that the S′ pockets within the active site are likely the second binding site for substrate, we reasoned that although the V50F-M51F mutant cannot be hydrolyzed as efficiently as WT, it should still be effectively bound by γ-secretase in an initial docking site. Co-IP of WT and V50F-M51F C100 in complex with γ-secretase reveals equal amounts of substrate bound, suggesting both substrates have a similar binding affinity for the enzyme (*Figure 5C*). Furthermore, myc-tagged V50F-M51F was just as effective as myc-tagged WT C100 at competing for γ-secretase cleavage of FLAG-tagged WT C100 (*Figure 5D*), again indicating V50F-M51F and WT substrate initially interact with γ-secretase in a similar manner. Together this suggests that the two Phe mutations in V50F-M51F do not affect initial recognition or binding of the C100 substrate, but rather may reduce the stabilization of a cleavable intermediate enzyme-substrate complex by clashing with the S′ pockets in the presenilin active site.

We noticed that in addition to being processed more slowly, V50F-M51F also produced a sharply lower Aβ42/40 ratio compared to WT (*Figure 5E*). To determine why this was occurring, we performed MS on the AICD fragment. Surprisingly, we found that ε cleavage was initiated almost exclusively after I47 (the very next available cleavage site), generating a 48–99 AICD fragment (*Figure 5F*), indicating γ-secretase is capable of skipping the two phenylalanines to initiate cleavage, albeit at a slower rate. Aβ47 is primarily processed to Aβ40 (Aβ47 → Aβ43 → Aβ40) (*Matsumura et al., 2014*), accounting for the decreased Aβ42/40 ratio.

Although γ-secretase usually cleaves APP in increments of three amino acids to produce predominantly Aβ40 and Aβ42 in the mechanism outlined in this study, a seemingly unusual cleavage deviates from the tripeptide preference to produce Aβ38 in appreciable quantities (*Okochi et al., 2013*; *Takami et al., 2009*). To determine if production of Aβ38 occurs through the same three S′ pockets used by γ-secretase to achieve tripeptide cleavage, we transiently transfected HEK cells with mutant APP containing a single Phe point mutant at V39, V40, I41 or A42. Measuring secreted Aβ38

**Figure 4.** Phenylalanine mutations in the P2' position of the last read tripeptide segment dictates final pathway preference. (**A**) Aβ42/40 ratios from HEK cells of V44F-M51F and I47F-M51F double mutants behave like single Phe mutants V44F and I47F, respectively. Aβ levels measured by 4G8 ELISA. Mean ± SD, n = 3, t-test **<0.01, ***<0.001. (**B**) Aβ42/40 ratios from HEK cells of I45F-V50F and T48F-V50F double mutants behave like single Phe mutants I45F and T48F, respectively. Aβ levels measured by 4G8 ELISA. Mean ± SD, n = 3, t-test *<0.05, **<0.01, ***<0.001, ****<0.0001. (**C**) MALDI/TOF MS conformation of the elimination of AICD 49–99 and AICD 50–99 for the V50F and M51F containing double Phe mutants, respectively. V44F-M51F (AICD 49–99, expected mass: 7034.8, observed mass: 7030.7; AICD 47–99 expected mass: 7249.0, observed mass: 7253.7), I45F-V50F (AICD 50–99, expected mass: 6953.8, observed mass: 6950.1; AICD 48–99, expected mass: 7167.9, observed mass: 7164.6), I47F-M51F (AICD 49–99, expected mass: 7034.8, observed mass: 7032.2; AICD 47–99, expected mass: 7283.0, observed mass: 7280.1), T48F-V50F (AICD 50–99, expected mass: 6953.8, observed mass: 6949.4; AICD 48–99, expected mass: 7214.0, observed mass: 7209.4). (**D**) Aβ42/40 ratios from HEK cells transfected with double Phe mutations in tandem. Aβ levels measured by 6E10 ELISA. Mean ± SD, n = 3, t-test **<0.01, ***<0.001, ****<0.0001.



**Figure 5.** Phenylalanine blocking mutations at both ε cleavage sites reduces APP cleavage but not binding to γ-secretase. (**A**) Western blot of γ-secretase cleavage of WT, V50F, M51F and V50F-M51F C100-FLAG. Duplicates from each substrate represent separate independent data points. * denotes a degradation product which co-purified with the substrate. (**B**) Cleavage of WT and V50F-M51F C100-FLAG over time. (**C**) Co-immunoprecipitation of Myc-tagged WT or V50F-M51F C100 substrate. Duplicates are from separate pull-down experiments. * antibody light chain. (**D**) Competitive cleavage of WT C100-FLAG by WT C100-Myc or V50F-M51F C100-Myc. (**E**) Aβ42/40 ratio of the V50F-M51F double mutant. Mean ± SD, n = 3, t-test, ****<0.0001. (**F**) MALDI/TOF MS of the AICD fragment from the V50F-M51F mutant: (AICD 51–99, expected mass: 6822.5, observed mass: 6817.1; * unknown peak, observed mass: 7030.4; AICD 48–99, expected mass: 7183.9, observed mass: 7179.2; AICD 47–99, expected mass: 7297.1, observed mass: 7292.5, AICD 46–99, expected mass: 7396.2, observed mass: 7396.1).

revealed that V40F almost completely eliminated Aβ38 production (*Figure 6A*), demonstrating occupancy of the three S' pockets is required for Aβ38 formation. V39F and I41F both sharply increased Aβ38, a result we interpret as the phenylalanine mutations making the precursor to Aβ38 a better substrate for γ-secretase, as the S1' and S3' pockets are large and prefer phenylalanine over smaller



**Figure 6.** γ-Secretase preferentially cleaves APP near the helix-destabilizing Gly-Gly motif. (**A**) Aβ38 levels from HEK cells transiently transfected with V39F, V40F, I41F or A42F APP. Aβ levels measured by 4G8 ELISA. Mean ± SD, n = 3. (**B**) Aβ42/40 ratios from V44F-I47F-V50F and I45F-T48F-M51F triple mutants from transiently transfected HEK cells. Aβ levels measured by 4G8 ELISA. Mean ± SD, n = 3, t-test, ****<0.0001. (**C**) Aβ38, Aβ40 and Aβ42 levels from HEK cells transfected with V44F-I47F-V50F and I45F-T48F-M51F triple mutants, the V44F-I45F double mutant and the hexa-mutant V44F-I45F-I47F-T48F-V50F-M51F. Aβ levels measured by 4G8 ELISA. Mean ± SD, n = 3. (**D**) Schematic diagram of sequential Phe mutants in the TMD of APP. (**E**) Aβ38 + 40 + 42 secreted from HEK cells transiently transfected with the mutants from (**D**). Aβ levels measured by 4G8 ELISA. Mean ± SD, n = 3, t-test ***<0.001, ****<0.0001. (**F**) Aβ42/40 ratios from (**E**). Aβ levels measured by 4G8 ELISA. Mean ± SD, n = 3, t-test *<0.05, ****<0.0001.

*Figure 6 continued on next page*

*Figure 6 continued*

The following figure supplement is available for figure 6:

**Figure supplement 1.** HEK cell expression and Aβ production from consecutive phenylalanine APP mutants.

amino acids (*Esler et al., 2004*). A42F produces about as much Aβ38 as WT, indicative of the lack of a contributory S4' pocket.

During the course of this study, we found that three phenylalanines sequentially mutated in the P2' positions of each major pathway (V44F-I47F-V50F and I45F-T48F-M51F) caused a very strong reduction or elevation in the Aβ42/40 ratio in the predicted direction (*Figure 6B*). As expected, the shifts were caused by the near complete elimination of Aβ42 for the V44F-I47F-V50F mutant and of Aβ40 for the I45F-T48F-M51F mutant (*Figure 6C*). Interestingly, in each case, Aβ38 levels were produced in amounts comparable to or greater than WT. This demonstrates that Aβ38 is capable of being produced from both the Aβ49 → 40 and Aβ48 → 42 pathways, in perfect agreement with recent MS studies that identified the precursors to Aβ38 as being either Aβ43 or Aβ42 (*Matsumura et al., 2014*; *Olsson et al., 2014*). Surprisingly, when we blocked the production of both Aβ43 and Aβ42 at the same time with a V44F-I45F double mutant we not only did not prevent the production of Aβ38, but rather Aβ38 levels were drastically increased (*Figure 6C*). Even after blocking the first six major cleavage sites (V44F-I45F-I47F-T48F-V50F-M51F) of APP, we still observe elevated Aβ38 compared to WT, demonstrating γ-secretase is fully capable of traversing multiple phenylalanines within APP to find the especially labile amide bond between G38 and V39. It is likely that the helix-destabilizing Gly-Gly motif at G37 and G38 is the reason for this observation, making the G38-V39 bond particularly accessible for cleavage.

Next, we attempted to determine how many phenylalanines in a row γ-secretase was capable of skipping by taking advantage of the fact that the GG motif apparently allows for γ-secretase to deviate from normal sequential tripeptide cleavage. Astonishingly, even after increasing the number of phenylalanines in a row to eight, γ-secretase was still able to produce Aβ above mock transfected levels (*Figure 6D,E*). The V44F⇒M51F mutant produced mostly Aβ38 (*Figure 6—figure supplement 1*), in what may be a single endoproteolytic cleavage event, although we have not been able to obtain enough AICD for MS confirmation. Predictably, the Aβ42/40 ratios for these mutants follow a pattern expected if γ-secretase cleaves at the next available site after each additional Phe (*Figure 6F*).

Using our newfound knowledge of the basic cleavage mechanism of γ-secretase, and our ability to precisely control it with phenylalanine mutations, we next decided to investigate the mechanism of FAD mutations within the TMD of APP, which all increase the Aβ42/40 ratio to different degrees. There are more than a dozen missense FAD mutations targeting this region of APP (*Figure 1A*), with the majority being located N-terminal and downstream of ε cleavage. To date, the currently accepted explanations for how these mutations increase the Aβ42/40 ratio are: 1) affecting the positioning/helical stability of the γ-secretase bound APP TMD such that initial ε cleavage is shifted toward T48, thus favoring the production of Aβ42 (*Chávez-Gutiérrez et al., 2012*; *Chen et al., 2014*; *Dimitrov et al., 2013*); or 2) influencing the reaction kinetics of γ-secretase's processing of APP, leading to incomplete carboxy-trimming and therefore increased Aβ42 over the more processed Aβ38 (*Chávez-Gutiérrez et al., 2012*) (*Figure 7A*). As demonstrated above with the I45F FAD mutant, we now show that a third possible mechanism exists, in that sequence-specific cleavage preferences can uncouple initial ε cleavage and final γ cleavages of APP (*Figure 7A*).

To determine the prevalence of the two previously proposed mechanisms and to possibly identify additional pathway uncoupling mutants, we made each FAD mutant within the APP TMD alone and as a double mutant with V50F. The V50F mutation would be predicted to block ε cleavage after T48 independent of the FAD mutant's affect on subsequent cleavage events. We would therefore predict that if an FAD mutant causes an increase in the Aβ42/40 ratio by influencing initial ε cleavage or by affecting subsequent carboxy-trimming along the Aβ48 → 42 pathway, then that same FAD mutant when paired with V50F should produce a reduced Aβ42/40 ratio compared to WT, similar to V50F alone. Conversely, if an FAD mutant affects subsequent cleavage events independent of ε cleavage,

**Figure 7.** The tripeptide cleavage mechanism of γ-secretase and the effect of APP transmembrane domain FAD mutations. (**A**) The three mechanisms by which FAD mutations within the TMD of APP increase the Aβ42/40 ratio. 1) Mutations shift initial ε cleavage towards the Aβ42 pathway. 2) Mutations reduce cleavage of the third cleavage event, producing more Aβ42 over the more processed Ab38. 3) Cleavage specific preferences cause switching from the Aβ40 to the Ab42 pathway, as exemplified by the I45F FAD mutant. (**B**) The Aβ42/40 ratio of each FAD mutation with and without an additional V50F mutation to control the ε cleavage site. The majority of mutations are rescued by the V50F substitution suggesting that these FAD mutations increase the Aβ42/40 ratio by influencing ε cleavage and/or affecting carboxy-trimming. I45F, I45T and T48P retain significantly elevated ratios, indicating these mutants dissociate initial ε and final γ cleavages. Aβ levels measured by 4G8 ELISA. Mean ± SD, n = 3, t-test ***<0.001, ****<0.0001.(**C**) The tripeptide cleavage mechanism of γ-secretase. After initial substrate binding, we speculate that the helical TMD of substrate unwinds into the active site of presenilin (PSEN) where it is

*Figure 7 continued on next page*

*Figure 7 continued*

stabilized by the three S' pockets in the catalytic pocket prior to cleavage. Successive carboxy tripeptide trimming occurs until the eventual release of Aβ peptide. (**D**) γ-secretase cleavage of the transmembrane domain of Notch from *Okochi et al, 2002*.

The following figure supplements are available for figure 7:

**Figure supplement 1.** APP FAD mutant panel measured by 6E10 ELISA.

**Figure supplement 2.** AICD fragments for the three I45 FAD mutants determined by western blot using the AICD 50–99 specific antibody.

**Figure supplement 3.** Secreted Aβ levels from V40F, A42F and V44F.

causing uncoupling of initial ε cleavage and final γ cleavages, then the FAD-V50F double mutant should retain an elevated Aβ42/40 ratio, similar to the FAD mutant alone.

Screening nearly all the FAD mutants within the TMD of APP by this method, we found that the majority were either completely or nearly completely rescued when paired with V50F (*Figure 7B*, *Figure 7—figure supplement 1*), resulting in 42/40 ratios significantly less than WT. This suggests the predominant mechanism of elevating the Aβ42/40 ratio by FAD mutants within the TMD of APP is by shifting the preference of initial ε cleavage from the Aβ40 to the Aβ42 pathway, and/or by influencing carboxy-trimming. This may have been predicted, given that I45F is the only aromatic amino acid mutation to fall in the S2' pocket of one of the major cleavage pathways. The only other FAD mutation containing an aromatic amino acid, V46F, falls within the S1' and S3' pockets for the Aβ42 and Aβ40 pathways, respectively, therefore never clashing with the S2' pocket and not influencing the Aβ42/40 ratio as a major pathway blocker. However, we found that in addition to I45F, there are two other mutants, I45T and T48P, which appear to dissociate the normal connection between initial pathway preference and final cleavage products.

V50F partially rescues the Aβ42/40 ratio when paired with I45T, but remains significantly elevated compared to WT, indicating the I45T mutant both influences initial ε cleavage and uncouples ε from γ cleavages. The change in ε cleavage preference of I45T was verified using the AICD 50–99 specific antibody (*Figure 7—figure supplement 2*), showing a small reduction in AICD 50–99. Interestingly, the I45T mutant reduces the amount of AICD 50–99 comparable to I45V, even though these two mutants display very different Aβ42/40 ratios. This again suggests I45T dissociates cleavage downstream of ε to achieve such a high Aβ42/40 ratio. Exactly how I45T does this is currently unknown and requires further investigation.

The T48P-V50F double mutant behaves identically to T48P alone. Given that proline is a helix-breaking amino acid, the T48P mutant may not undergo normal ε cleavage after P48 or L49. Determining how T48P increases the 42/40 ratio and overcomes the ε controlling V50F mutant will require further investigation.

## Discussion

In this study, we identify that three S' amino acid binding pockets guide the productive positioning of substrate into the γ-secretase active site, providing the mechanism behind the enzyme's preferred tripeptide cleavage of APP and pathogenic Aβ production. Based on the data reported herein (discussed further below) and numerous other studies of γ-secretase (*Das et al., 2003*; *Kornilova et al., 2003*, *2005*) and other intramembrane proteases (*Akiyama et al., 2015*; *Dickey et al., 2013*; *Fluhrer et al., 2012*; *Moin and Urban, 2012*; *Urban and Freeman, 2003*; *Ye et al., 2000*), we speculate that after initial binding to γ-secretase, substrate must undergo a translocation and/or conformational change in order to bind the three S' pockets within the active site and subsequently be cleaved by the enzyme (*Figure 7C*). After initial endoproteolysis, the three S' pockets guide further carboxy-trimming of the retained Aβ species until it is short enough to dissociate from the complex, producing predominantly Aβ38, Aβ40 and Aβ42.

More than a decade ago, SAR studies of γ-secretase-targeting transition-state analogs putatively assigned three S' pockets to the active site of the enzyme (*Esler et al., 2004*). Transition-state analogs containing only two amino acids for S' pocket binding inhibited γ-secretase less effectively, indicating that occupancy of the S3' pocket is required for a strong interaction. This likely provides the reason why γ-secretase only cleaves APP in segments of three or more amino acids, but never two. The preference for cleaving three amino acids originates from the lack of a contributory fourth S' pocket. A fourth amino acid (no matter the size) added to a transition state inhibitor neither increased nor decreased the inhibitors potency (*Esler et al., 2004*), suggesting a fourth S' pocket doesn't contribute to γ-secretase-inhibitor or -substrate interactions.

To support our proposed model, we exploited the fact that the second of the three S' pockets is apparently too small to readily accommodate an aromatic amino acid. We generated dozens of substrates containing aromatic amino acid substitutions in the P2' positions at each cleavage site along the Aβ40 or Aβ42 pathways, selectively blocking each individual cleavage event. Without exception we were able to predict the shift in the Aβ42/40 ratio. This was accomplished both in vitro with purified γ-secretase and recombinant C100 substrate as well as in a cell-based assay, transiently transfecting mutant full-length APP in HEK cells and measuring secreted Aβ. The same predicted Aβ42/40 ratio changes were observed in vitro and in the cell-based assay whether we measured Aβ (6E10 detection antibody) or Aβ plus p3 products (4G8 detection antibody) by ELISA. Together this demonstrates we are probing the fundamental mechanism by which γ-secretase cleaves APP, irrespective of mutant effects on cellular localization, γ-secretase's interaction with activity-modulating proteins/lipids within the cell or any artifacts that may arise from more artificial in vitro assays.

Although we are unable to measure every cleavage product from the ~70 mutant forms of APP we generated, we do note that every time a Phe was placed in the P2' position of a cleavage product that we could readily and directly measure, there were almost negligible amounts of that product formed. For example, in the cell-based assay, V40F, A42F and V44F generated levels of Aβ38, Aβ40 and Aβ42, respectively, that were actually less than mock transfected levels (*Figure 7—figure supplement 3*). These Aβ levels are orders of magnitude less than that from WT APP transfected cells, although we cannot say whether these low levels of Aβ species were produced from endogenous HEK cell APP or from our transfected mutants. Similarly, we were unable to detect by MS any AICD products containing a Phe in the P2' position. Together, these data suggest that aromatic amino acids may be completely excluded from the S2' pocket and further demonstrates that substrate occupancy of the three S' pockets is an absolute requirement for catalysis.

It is likely that phenyalanine substitutions at various positions along the transmembrane domain of APP influence the general structure and/or helical stability of the substrate. This could affect the manner in which these substrates interact with γ-secretase. Given that we are able to accurately predict the Aβ42/40 ratio without exception for the dozens of mutants used in this study, we expect that cleavage preferences dictated by the presence of aromatic amino acids in the P2' position are overriding any affect these mutations have on substrate helical structure/stability and any altered manner in which these mutant substrates initially interact with the enzyme. This is directly supported in *Figure 3* where V44F, I45F, I47F and T48F all shift initial ε cleavage in favor of the opposite pathway relative to the final cleavage products measured and originally predicted by our model.

We present several lines of evidence suggesting that substrate movement and/or a substrate conformational change after initial enzyme binding is an important step in γ-secretase's catalytic mechanism. We identify that like transition-state analogs, tripeptide cleavage products are noncompetitive inhibitors of γ-secretase, albeit very weak inhibitors. By definition this means the binding sites on γ-secretase for initial substrate binding and subsequent catalysis are spatially separate, requiring substrate movement after initial binding to be an integral part of γ-secretase's catalytic mechanism. For rhomboid, a recent study demonstrates that product-mimicking peptide aldehydes are non-competitive inhibitors of this serine intramembrane protease (*Cho et al., 2016*), exactly like tripeptide products and transition-state analog inhibitors are for γ-secretase, suggesting a common two-step mechanism between these two intramembrane proteases.

Additionally, the double Phe mutant V50F-M51F, which is predicted by our model to sterically clash with the three S' pockets in the active site, is still efficiently bound to γ-secretase even though it is processed less efficiently than WT. Given that this mutant effectively competes for γ-secretase processing of other substrates, it must bind to the initial docking site on the enzyme for substrate.

This suggests that binding to the S' pockets is the final step in substrate recognition and positioning within the enzyme prior to catalysis.

Furthermore, we are unable to prevent the formation of Aβ38 by specifically blocking the production of the known Aβ38 precursors Aβ42 and Aβ43; instead, paradoxically the V44F-I45F double mutant increases Aβ38 production. It is likely that the helix-destabilizing Gly-Gly motif at G37 and G38 is the reason for this observation. Local helical unwinding around this position probably makes the amide bond between G38 and V39 particularly accessible for cleavage, and this may be the reason for γ-secretase's normal deviation from the preferred tripeptide cleavage for Aβ38 production. However, further investigation, including quantification of the intramembrane peptide products by mass spectrometry, will be required to prove this. In the absence of a γ-secretase—substrate co-complex structure, it will not be possible to definitively prove the existence of a partially unwound substrate intermediate. Conceivably, it should be possible to capture such an intermediate with a transition-state analog covalently linked to the C-terminus of an APP- or notch-based helical substrate.

Given that tripeptide cleavage is dictated by the three S' pockets in the active site of presenilin, we expect that other γ-secretase substrates will be similarly cleaved preferentially in increments of three amino acids, while skipping aromatic amino acids that fall in the S2' pocket along the way, and with cleavage occurring preferentially in helix-destabilized regions. This is important to note given that many of γ-secretase's substrates naturally contain aromatic amino acids. The TMD of notch, for example, naturally contains three phenylalanines. We know from a previous MS study that these phenylalanines are skipped by γ-secretase in a pattern consistent with our model (*Figure 7D*) (*Okochi et al., 2002*).

In a recent study, we have demonstrated that substrates with large ectodomains have a reduced binding affinity for γ-secretase due to steric clashing with the nicastrin component of the γ-secretase complex (*Bolduc et al., 2016*). Based on this, and results from our current study, we might expect that γ-secretase can chose its substrates through a complex interplay between ectodomain length, helical TMD stability and the sequence of amino acids (specifically aromatic amino acids) within substrate TMD. There may even exist non-substrates containing short ectodomains but having stable helices that are further protected from cleavage by sequential stretches of aromatic amino acids. Whether such non-substrates exist, and the relationship between ectodomain size, helical stability and amino acid sequence will require further investigation.

We show that the majority of FAD mutants within the TMD of APP primarily increase the Aβ42/40 ratio by changing ε cleavage, favoring the Aβ48 → 42 pathway. This is mostly in agreement with previous studies (*Chávez-Gutiérrez et al., 2012*; *Chen et al., 2014*; *Dimitrov et al., 2013*; *Quintero-Monzon et al., 2011*). However, we also identify a new mechanism by which certain FAD mutations can increase the production of pathogenic Aβ species. Here, final γ cleavages are uncoupled from initial pathway preference determined by ε cleavage. In the case of the I45F FAD mutation, the bulky Phe sterically clashes with the S2' pocket of γ-secretase at the Aβ43 cleavage site, blocking its cleavage and the subsequent production of Aβ40. It is likely that the positioning of the Phe in the S3' pocket of the Aβ42 cleavage also enhances γ-secretase proteolysis at this position through a favorable S3'-P3' interaction. These two interactions likely combine to account for the fact that no other mutation within the TMD of APP produces as much Aβ42 as I45F. This is the most severe APP FAD mutation, with an onset of clinical AD at 31 years of age (*Guerreiro et al., 2010*). There are at least two additional FAD mutations, I45T and T48P, that appear to dissociate the ε and γ cleavages. How these mutations accomplish this is currently unknown.

Our data also help explain previous observations in the literature. Prior to the identification of presenilin as being the protease responsible for γ-secretase activity (*Wolfe et al., 1999*), Lichtenthaler et al. performed a phenylalanine scanning study of the TMD of APP. In their study, they observed the exact same Aβ42/40 shifts identified here for several of the same mutants (*Lichtenthaler et al., 1999*). Later, Sato et al. found that γ-secretase was unable to cleave through stretches of 3–5 consecutive tryptophans inserted into the TMD of APP (*Sato et al., 2005*). These observations are now explained by the determination that the small S2' pocket of γ-secretase cannot accommodate aromatic amino acids.

With the identification of the key role the three S' pockets play in γ-secretase's cleavage mechanism, several important new questions are raised. The exact locations of the three S' pockets in presenilin are currently unknown. Additionally, the identity of the initial substrate-binding site on

presenilin is unknown. The resolution of enzyme—substrate and/or -inhibitor co-complexes by cryo-EM will be informative in this regard. All enzymatic assays in this study utilized purified γ-secretase complex containing presenilin-1. It will be interesting to see if presenilin-2 has similar substrate cleavage preferences. A major unsolved question pertains to the mechanism by which substrate TMD is repositioned within γ-secretase after each cleavage event in order to be close enough to the active site for the next round of catalysis to occur. Is this a ratcheting or sliding motion? Is this an active process or based on Brownian motion? At present, we do not know how γ-secretase is capable of skipping stretches of several phenylalanines in a row. Determining how γ-secretase accomplishes this may help elucidate how substrate moves during normal sequential cleavage.

Now that we have identified several key aspects of γ-secretase's substrate recognition and cleavage mechanisms, as well as provide valuable new tools for future structural and biochemical studies, we should be able design experiments to elucidate some of γ-secretase's remaining unanswered functional questions. The answers from which should have further broad implications for our understanding of Alzheimer's disease and the development of safe and effective therapeutics targeting γ-secretase function.

# Materials and methods

## Materials

The following antibodies were used: α-Myc (9E10 Santa Cruz #sc-40), α-Flag (M2 Sigma #F3165), α-Nct (Cell Signaling 3632S), α-GAPDH (Cell Signaling Technology, ab125247), α-APP (C7), α-AICD 50–99 (Rb) was a kind gift from Philip Szekeres at Eli Lilly, α-Ms 800nm (Licor Bio 926–32212) and α-Rb 680nm (Licor Bio 926–68021). Tripeptides were synthesized by Anaspec corp. Total brain lipid extract was from Avanti Polar Lipids (#131101). The following Aβ ELISA kits were used: 4G8 (Meso Scale Diagnostics, K15199E) or 6E10 (Meso Scale Diagnostics, K15200E) for cell-based assays; Aβ40 (#KHB3482) and Aβ42 (#KHB2442) ELISA kits from Invitrogen for in vitro assays.

## Cloning

All mutant forms of C100 FLAG or full-length APP were generated by site directed mutagenesis of either C100-FLAG in pET22b or full-length WT APP in the pCMV695 plasmid.

## Tissue culture and transfection of adherent cells

Adherent HEK cells were cultured in complete growth media: Dulbecco's Modified Eagle's Medium (DMEM) supplemented with 10% fetal bovine serum (FBS), 2 mM L-glutamine, 10 Units/mL penicillin, and 10 μg/mL streptomycin. For transfection, adherent HEK cells were seeded in six-well dishes at a density of $5\times10^5$ cells per well. Transfection was carried out with Lipofectamine 3000 reagent in serum-free conditions with Opti-MEM I. Cells were incubated for 24 hr, at which time conditioned media was harvested for ELISA and cells were harvested for western blot.

## Growth and purification of γ-secretase from HEK cells

Suspension HEK cells were cultured in 100 mL of unsupplemented Freestyle 293 media (Life Technologies, 12338-018) with shaking at 125 rpm, and passaged at a density of $2 \times 10^6$ cells/mL. For transfection, suspension HEK cultures were grown to a density of $2 \times 10^6$ cells/mL. Media was replaced with fresh Freestyle 293 media. 5 mL Freestyle 293, 150 μg of γ-secretase vector containing presenilin 1 (provided by Yigong Shi), and 450 μg of 25-kDa linear polyethylenimines (PEI) was mixed and incubated for 30 min at room temperature. The DNA/PEI solution was then added to the HEK culture and cells were grown for ~60 hrs prior to harvesting. γ-secretase was purified as previously described (*Fraering et al., 2004*; *Osenkowski et al., 2009*)

## Purification of C100-FLAG substrates

C100-FLAG substrates were expressed in BL21 *E. coli* for 3 hrs at 37°C after induction with 1 mM IPTG. Cells were then pelleted and lysed by French press in 50 mM HEPES pH 7.0, 1% Triton X-100 detergent. FLAG-tagged substrates were then isolated by immunoprecipitation for 3 hrs at 4°C with anti-FLAG M2 beads from Sigma. Substrates were then eluted from the beads with 100 mM glycine pH 2.5, 0.25% NP40 prior to being neutralized with tris buffer and stored at -80°C.

### In vitro γ-secretase assay

Purified γ-secretase was incorporated into vesicles by first dissolving total brain lipid extract (1.25 mM final) in 50 mM HEPES pH 7.0, 150 mM NaCl, 0.25% CHAPSO. γ-Secretase (5–30 nM final concentration) was then added to the solution and detergent removed by mixing SM-2 biobeads (62 mg/mL) (Bio-Rad) with the lipid/detergent/enzyme solution for two hrs at 4°C. Biobeads were removed from the newly formed proteoliposomes and reactions were initiated with the addition of purified recombinant substrate C100-FLAG substrate. Reactions were quenched with SDS loading dye for western blot or centrifuged for ELISA or mass spectrometry on Aβ or AICD fragments, respectively.

For inhibition studies, tripeptide fragments or inhibitors were dissolved in DMSO prior to being diluted into the reaction buffer. The concentration of C100-FLAG used in all in vitro assays was 500 nM unless otherwise stated.

### Aβ ELISA

Conditioned media from HEK cells transfected with WT or mutant APP was assayed for Aβ by 4G8 or 6E10 Aβ ELISA kits from Meso Scale Diagnostics. Aβ levels were measured by both 6E10 and 4G8 ELISA for each data point, yielding nearly identical results. In vitro assay Aβ was measured using Aβ40 and Aβ42 ELISA kits from Invitrogen. All ELISAs were performed according to the manufacture's protocols.

### Co-immunoprecipitation of purified γ-secretase and substrate

Purified γ-secretase (5 nM final concentration) was preincubated in assay buffer (50 mM HEPES pH 7.0, 150 mM NaCl, 0.25% CHAPSO, 0.1% DOPC and 0.025% DOPE, 2% BSA) in the presence of 2 µM III-31C for 1 hr at room temperature. Purified WT or mutant C100-Myc (20 nM final concentration) was then incubated with γ-secretase for 1 hr prior to pull down with anti-HA magnetic affinity beads for 4 hr with mixing at room temperature. The immunoprecipitated complex was then washed three times and eluted with SDS loading buffer prior to western blot with an anti-Myc antibody.

### Kinetic analysis

Tripeptide $IC_{50}$ inhibition was fit to:

$$\frac{v_i}{v_o} = 1 / \left(1 + \frac{[I]}{IC_{50}}\right)$$

where $v_i$ is the initial velocity in the presence of inhibitor at concentration $[I]$ and $v_o$ is the initial velocity in the absence of inhibitor.

Tripeptide inhibition was globally fit to the following noncompetitive equation:

$$v = \frac{V_{max}[S]}{[S]\left(1 + \frac{[I]}{K_{ii}}\right) + K_m\left(1 + \frac{[I]}{K_i}\right)}$$

where, $v$ is the initial rate, $K_i$ is dissociation constant for inhibitor binding to free enzyme, $K_{ii}$ is the dissociation constant for inhibitor binding to the enzyme-substrate complex.

Inhibitor cross-competition was globally fit to:

$$\frac{1}{v_{ij}} = 1 / v_0 \left(1 + \frac{[I]}{K_i} + \frac{[J]}{K_j} + \frac{[I][J]}{\alpha K_i K_j}\right)$$

where, $v_{ij}$ is the initial rate in the presence of inhibitors, $v_0$ is the initial rate in the absence of inhibitor, $K_i$ and $K_j$ are the dissociation constants for inhibitors $I$ and $J$, respectively. α = ∞ for inhibitors which bind in a mutually exclusive fashion, while α = 1 for inhibitors which have distinct binding sites.

### Total Aβ ELISA

Conditioned media was collected from transiently-transfected adherent HEK cells. Each well of an uncoated 96-well multi-array plate (Meso Scale Discovery, #L15XA-3) was coated with 30 µL of a PBS solution containing 3 µg/mL of 266 capture antibody (Elan), and incubated at room temperature

overnight. A detection antibody solution was prepared with 3D6B detection antibody (Elan), 100 ng/mL Streptavidin Sulfo-TAG (Meso Scale Discovery, #R32AD-5), and 1% MSD Blocker A (#R93BA-4) in wash buffer (#R61TX-1). Following overnight incubation, 25 μL/well of sample, followed by 25 μL/well of detection antibody solution were incubated for 2 hr at room temperature with shaking at >300 rpm, washing wells with wash buffer between incubations. Plate was read and analyzed according to manufacturer protocol.

### Immunoprecipitation-Mass spectrometry (IP-MS)

Following an in vitro proteoliposome activity assay, AICD-FLAG products were isolated by immuno-precipitiation with anti-FLAG M2 magnetic beads from Sigma. Completed reactions were incubated with 50 μL of M2 beads in 10 mM MES pH 6.5, 10 mM NaCl, 0.05% DDM detergent in 500 μL volumes overnight at 4°C. AICD was then eluted from the beads with acetonitrile:water (1:1) with 0.1% trifluoroacetic acid. MALDI/TOF mass spectrometry was performed with sinapinic acid matrix on a calibrated ultraflextreme MALDI/TOF/TOF from Bruker in linear mode.

## Acknowledgements

This work was funded by the following NIH grants: R01 AG06173 (DJS), program project grant AG015379 (MSW and DJS).

## Additional information

### Funding

| Funder | Grant reference number | Author |
| --- | --- | --- |
| National Institute on Aging | R01 AG06173 | David M Bolduc<br>Daniel R Montagna<br>Matthew C Seghers<br>Michael S Wolfe<br>Dennis J Selkoe |
| National Institute on Aging | AG015379 | David M Bolduc<br>Daniel R Montagna<br>Matthew C Seghers<br>Michael S Wolfe<br>Dennis J Selkoe |

The funders had no role in study design, data collection and interpretation, or the decision to submit the work for publication.

### Author contributions

DMB, Conception and design, Acquisition of data, Analysis and interpretation of data, Drafting or revising the article; DRM, Conception and design, Acquisition of data, Analysis and interpretation of data; MCS, Acquisition of data, Analysis and interpretation of data; MSW, DJS, Conception and design, Analysis and interpretation of data, Drafting or revising the article

### Author ORCIDs

David M Bolduc, http://orcid.org/0000-0002-2000-5728
Matthew C Seghers, http://orcid.org/0000-0002-2446-7102

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
