## [Decision Letter]

Thank you for submitting your article "The amyloid-beta forming tripeptide cleavage mechanism of γ-secretase" for consideration by *eLife*. Your article has been reviewed by Sin Urban (Reviewer #1) and Charles R Sanders (Reviewer #2), and the evaluation has been overseen by a Reviewing Editor and Randy Schekman as the Senior Editor.

The reviewers have discussed the reviews with one another and the Reviewing Editor has drafted this decision to help you prepare a revised submission.

Summary:

The reviewers and Reviewing Editor were in strong agreement that your manuscript reports a compelling set of findings that have important basic and translational implications for the field of Alzheimer's research and the role that γ-secretase-mediated APP processing plays in this disease. The reviewers have made some specific suggestions for revision that are summarized below.

Essential revisions:

*Reviewer 1:*

1) The generation of Aβ38 does not obviously fit the tripeptide assembly line mechanism. Based on their poly-phenylalanine mutants, the authors suggest that Aβ38 is generated by a jump over the normal tripeptide assembly line cleavage sites as afforded by the di-glycine motif serving as a helix break. This is a very attractive hypothesis, but unfortunately it is not tested. The authors should check whether mutating the di-glycine motif to di-alanine (or alanine-valine) does not disrupt the normal assembly line processing, but blocks the jump to generate Aβ38. This experiment is feasible with the existing assays the authors have developed.

2) In my opinion, one of the truly beautiful insights of the work is the uncoupling of the initiating versus processive cleavages by mutations. I nevertheless found it more difficult to follow the same logic when a similar experiment was introduced with FAD mutations in APP in Figure 7. For example, the effect of internal mutants was not affected by either V50F or M51F in Figure 4, but then in 7B the V50F mutant seemed to rescue the effect of most FAD mutations. I suspect one key difference is that most FAD mutants don't introduce F. But two did, and there seem to be differences in the result with I45F+/-V50F in Figure 7 (double mutant is elevated) versus the same mutants in 4B (seem not different from each other). Although I understand that the authors faced manuscript length challenges with this large body of work, I would suggest that they nevertheless spend a little more space explaining these effects when they appear at the end of the Results section.

*Reviewer 2:*

1) There are now high/medium resolution cryo-EM structures available for γ-secretase. The authors should comment on whether it is yet possible to tie their mechanistic results to the structure of γ-secretase. Is the active site in γ secretase seen to be located near the cytosolic end of the trasmembrane domain, as might be expected based on the location of the epsilon cleavage sites in C99 near the end of its TMD? Can candidate S1'-S2'-S3' sites be identified in γ-secretase next to the hydrolytic site? Is there an evident pre-Michaelis substrate docking site? Readers are going to wonder about these issues.

2) If the paper is revised, please add page numbers for the sake of the reviewers/editors.

3) For this study, I assume that the form of γ-secretase they are working with has a presenilin-1 subunit. The authors should specifically point this out and comment on whether presenilin-2 can be assured to operate based on the same principles of cleavage site specificity.

4) First page of Results: "oftentimes" not "often times"

5) Last paragraph before Discussion: "after P48" not "after T48"

6) There is a body of work involving studies of the structure and dynamics of C99. Some of this work (papers by Dieter Langosch, for example) has examined the intrinsic helical stability of different parts of the C99 TMD. It is not clear if the authors are aware of this work (see "in the absence of structural information" in Discussion), but they may want to consider whether there are any published results that do actually provide insight. Even if not, this may be worth commenting on.

7) The end of the Discussion provides a nice summary of their work. However, one part of their data that I think they should comment on again is the fact that γ-secretase can skip over substrate sequences that are tracts of Phe. Along with this is the related question of what (even for WT C99 and other substrates) is responsible for successive threading of the remaining part of the substrate into the active site following initial cleavage at epsilon and intermediate cleavage sites-is this likely to be an energetically-directed mechanism or is it simply based on the Brownian motion of the substrate already bound to the active site? Commenting on these matter even to point out where the current data does not inform may help to point to important future directions.

---

## [Author Response]

*Essential revisions:*

*Reviewer 1:*

*1) The generation of Aβ38 does not obviously fit the tripeptide assembly line mechanism. Based on their poly-phenylalanine mutants, the authors suggest that Aβ38 is generated by a jump over the normal tripeptide assembly line cleavage sites as afforded by the di-glycine motif serving as a helix break. This is a very attractive hypothesis, but unfortunately it is not tested. The authors should check whether mutating the di-glycine motif to di-alanine (or alanine-valine) does not disrupt the normal assembly line processing, but blocks the jump to generate Aβ38. This experiment is feasible with the existing assays the authors have developed.*

We fully agree that investigating the role of decreased helical stability around Gly37 and Gly38 in the sequential cleavage of the TMD of APP is a very important mechanistic question not fully answered by our study. However, we believe that substantial additional work would be required to fully answer this question, as we do not have assays in place to address this. In the current study, we rely heavily on antibodies that specifically recognize the neo-epitopes of Aβ38, Aβ40 and Aβ42 produced by γ-secretase cleavage of the TMD of APP. Here, G37 and G38 are present in the recognition epitope of all three antibodies. Therefore, mutating these amino acids abrogates binding and recognition by these antibodies. The only way to definitively determine if mutation of these two Gly residues causes γ-secretase to cleave a tripeptide instead of a tetrapeptide here is to measure by LC-MS the intramembrane tri- and tetra- cleavage fragments of APP. Earlier this year we initiated a collaboration with the Ihara group (Doshisha University, Japan) to obtain mass spec analysis on the TMD cleavage fragments of several APP mutants we have generated. They are the only group worldwide that has reported the ability to capture these highly hydrophobic fragments and quantify them by MS. This collaboration is ongoing and not yet complete. Thus, this is an important question that will likely require a follow-up study to answer completely. We are careful in this current manuscript to propose this mechanism as a possible explanation for tetra-peptide generation of Aβ38 that is consistent with all our other data, but not one we have definitively answered.

*2) In my opinion, one of the truly beautiful insights of the work is the uncoupling of the initiating versus processive cleavages by mutations. I nevertheless found it more difficult to follow the same logic when a similar experiment was introduced with FAD mutations in APP in Figure 7. For example, the effect of internal mutants was not affected by either V50F or M51F in Figure 4, but then in 7B the V50F mutant seemed to rescue the effect of most FAD mutations. I suspect one key difference is that most FAD mutants don't introduce F. But two did, and there seem to be differences in the result with I45F+/-V50F in Figure 7 (double mutant is elevated) versus the same mutants in 4B (seem not different from each other). Although I understand that the authors faced manuscript length challenges with this large body of work, I would suggest that they nevertheless spend a little more space explaining these effects when they appear at the end of the Results section.*

We have now attempted to better explain the logic behind the experiment in Figure 7 in the Results section (paragraphs 19-21). As the reviewer correctly surmises, the key reason V50F rescues the majority of internal TMD APP FAD mutants with respect to the Aβ42/40 ratio is because most of the FAD mutants are not a Phe (or other aromatic amino acid). There are two Phe FAD mutants, one at I45 and one at V46. The I45F mutant clashes with the small S2’ pocket in the Aβ40 pathway (therefore blocking this pathway and increasing the 42/40 ratio). In contrast, V46F falls in the S1’ and S3’ pockets for the 42 and 40 pathways, respectively, therefore never clashing with the S2’ pocket and not influencing the Aβ42/40 ratio as a major pathway blocker.

The only difference between Figure 7 and Figure 4 is that their data were analyzed with different ELISA kits. Figure 4 Aβ levels were measured with a 4G8 ELISA, which measures both β- and α-secretase generated products, while Figure 7 Aβ levels were measured with a 6E10 ELISA, which measures only β-secretase produced products. The 4G8 ELISA allowed for increased signal, given that it measures both α- and β-secretase cleavage products. With some mutants this increased signal was necessary to make reliable measurements (such as with the poly-Phe mutants, Figure 6). For the I45F-V50F data point in Figure 7, the amount of Aβ40 produced was quite low, near the limit of detection of the 6E10 ELISA, possibly inflating the Aβ42/40 ratio. We realize now this is confusing and regret omitting which ELISA detection kit we used in each figure legend (we have now done so), though we stated in the Methods section that we used both kits for each data set. To simplify this possible point of confusion for readers, we have replaced the 6E10 ELISA results in Figure 7 with the corresponding 4G8 results. There are no other differences here. To be thorough, we are now including the 6E10 data as an additional Figure 7—figure supplement 2, as well, so reviewers and readers can compare the results from the two ELISA kits side-by-side.

*Reviewer 2:*

*1) There are now high/medium resolution cryo-EM structures available for γ-secretase. The authors should comment on whether it is yet possible to tie their mechanistic results to the structure of γ-secretase. Is the active site in γ secretase seen to be located near the cytosolic end of the trasmembrane domain, as might be expected based on the location of the epsilon cleavage sites in C99 near the end of its TMD? Can candidate S1'-S2'-S3' sites be identified in γ-secretase next to the hydrolytic site? Is there an evident pre-Michaelis substrate docking site? Readers are going to wonder about these issues.*

The recent high-resolution structures of γ-secretase from Scheres and Shi undoubtedly provide a wealth of information for current and future mechanistic studies of γ-secretase. Indeed, the catalytic aspartates are near the cytosolic-membrane interface, as expected. We are unable to identify candidate S1’-S2’-S3’ pockets in the general vicinity of the catalytic aspartates at this time. The solution of transition-state analog – γ-secretase co-complexes by cryo-EM will likely identify these important substrate-binding pockets in the near future. Interestingly, the most recent cryo-EM structures of γ-secretase have revealed the presence of an unidentified molecule situated between TMDs 2, 3 and 5 of presenilin. Intriguingly there are numerous FAD mutations in these presenilin TMDs that surround this object. It is therefore conceivable that this position makes up the initial substrate-docking site on presenilin. The true identity of the initial docking site, however, will require solving a γ-secretase structure with a known substrate or helical inhibitor. We now briefly touch on this in the Discussion section (tenth paragraph).

*2) If the paper is revised, please add page numbers for the sake of the reviewers/editors.*

Page numbers have been added.

*3) For this study, I assume that the form of γ-secretase they are working with has a presenilin-1 subunit. The authors should specifically point this out and comment on whether presenilin-2 can be assured to operate based on the same principles of cleavage site specificity.*

The γ-secretase complex used in all in vitro assays in the study contains PS1 as its catalytic subunit. We believe it is likely that PS2 has similar cleavage site specificity, though we haven’t addressed this in the current study. We now briefly comment on this in the Discussion (thirteenth paragraph).

*4) First page of Results: "oftentimes" not "often times"*

Corrected.

*5) Last paragraph before Discussion: "after P48" not "after T48"*

Corrected.

*6) There is a body of work involving studies of the structure and dynamics of C99. Some of this work (papers by Dieter Langosch, for example) has examined the intrinsic helical stability of different parts of the C99 TMD. It is not clear if the authors are aware of this work (see "in the absence of structural information" in Discussion), but they may want to consider whether there are any published results that do actually provide insight. Even if not, this may be worth commenting on.*

Here in the Discussion, “in the absence of structural information”, we were referring to the lack of a co-complex structure between γ-secretase and substrate and/or inhibitor. We have clarified this in the revised manuscript.

There are several very nice studies examining the intrinsic helical stability and flexibility of the TMD of APP in solution. Though these studies provide valuable information regarding the structure and stability of APP, it is not possible to say what the conformation of the C99 substrate is when bound to γ-secretase without a co-complex structure between enzyme and substrate being solved. Our many Phe mutants certainly must influence the structure and stability of the APP TMD, but given that the 42/40 ratio of all of the Phe mutants were predicted by our model, we believe that the mutations’ effects on substrate structure and/or helical stability are either minor or more likely overridden by the cleavage specificity of the enzyme dictated by the S1’-S2’-S3’ pockets.

*7) The end of the Discussion provides a nice summary of their work. However, one part of their data that I think they should comment on again is the fact that γ-secretase can skip over substrate sequences that are tracts of Phe. Along with this is the related question of what (even for WT C99 and other substrates) is responsible for successive threading of the remaining part of the substrate into the active site following initial cleavage at epsilon and intermediate cleavage sites-is this likely to be an energetically-directed mechanism or is it simply based on the Brownian motion of the substrate already bound to the active site? Commenting on these matter even to point out where the current data does not inform may help to point to important future directions.*

We believe this is one of the most important unsolved questions regarding the basic cleavage mechanism of γ-secretase. Unfortunately, current data in the field (and in our study) offer little insight into this mechanism. We were quite surprised that γ-secretase was capable of skipping so many phenylalanines to initiate cleavage. We think these poly-Phe substrates may provide the tools required to begin interrogating this question and hopefully we will be able to contribute to the field’s understanding here in the future. We have commented briefly on this matter in the Discussion, as requested (thirteenth paragraph).